# Evaluation of Tires Acting on Soil in Field Conditions Using the 3D Scanning Method

**Weronika Ptak *** , **Jarosław Czarnecki, Marek Brennensthul** , **Krzysztof Lejman** and **Agata Małecka**

Institute of Agricultural Engineering, Wroclaw University of Environmental and Life Sciences,
51-630 Wroclaw, Poland; jaroslaw.czarnecki@upwr.edu.pl (J.C.); marek.brennensthul@upwr.edu.pl (M.B.);
krzysztof.lejman@upwr.edu.pl (K.L.); 113322@student.upwr.edu.pl (A.M.)
***** Correspondence: weronika.ptak@upwr.edu.pl

**Abstract:** This research presents the results of tests conducted under field conditions and included measuring the footprint of tires on soil. Two agricultural tires of the same size but different internal structures were tested, 500/50R17 (radial) and 500/50-17 (bias-ply). The factors were tire inflation pressure (0.8 bar, 1.6 bar, and 2.4 bar) and tire vertical load (7.8 kN, 11.8 kN, and 15.7 kN). The footprint made on the soil was scanned with a 3D scanner, resulting in a digital image of the tire footprint on the soil to enable an analysis of the measured parameters: length, width, depth, and contact area (in 3D form). Statistical analysis showed that for radial tire footprints, both inflation pressure and vertical load had a significant effect on all analyzed parameters. For bias-ply tire footprints, it was shown that only inflation pressure had a significant effect on all of the analyzed parameters, while the significance of the effect of the vertical load was not confirmed for the footprint depth. Based on the results obtained, the suitability of models describing the relationship between operating factors and the actual footprint area was verified. It was found that for a radial tire, the model formulated based on laboratory tests can predict the contact surface under field conditions (the correlation coefficient $R^2$ was equal to 0.9226). In the case of a bias-ply tire, the correlation coefficient $R^2$ reached a value equal to 0.5828. This indicates a less accurate estimation of the surface area under field conditions based on the model designed after laboratory testing.

**Keywords:** soil deformation; contact surface; radial tire; bias-ply tire; 3D scanning



## 1. Introduction

The main goal of agriculture is to produce food. This goal can be achieved using various agriculture systems, of which the intensive agriculture system is relatively popular. It is geared toward achieving maximum efficiency in agrotechnical measures [1,2]. Higher efficiency requires using machines with larger widths and the ability to develop higher operational speeds to treat as much area as possible in a given time. This, in turn, creates high power requirements for such machines—so it becomes necessary to use larger tractors. Unfortunately, there are negative side effects of such measures caused by their large weight. The market is responding very quickly to the needs of farmers, and today a trend can be observed—tractors in the middle and upper power range are the most popular, but at the same time, have higher weight than their counterparts of 20–30 years ago [3]. The high weight of field aggregates can cause soil compaction which harms the soil. As a result, the soil's water and air balance is disturbed, the volume of air pores in the soil decreases, rainwater absorption and plant root development are hindered, and the mechanical strength of the soil increases [4,5]. Thus, compacted and more resistant soil requires loosening, which involves energy-intensive procedures, and the fuel consumption of agricultural vehicles consequently increases [6]. For these reasons, there is a need to find solutions to reduce the negative effects of heavy agricultural equipment on the soil.

Soil compaction results from soil stresses caused by the interaction of the chassis system components of agricultural vehicles and machinery. The most popular running

gear in agricultural equipment is the wheeled chassis, which has the tire in direct contact with the soil. The tracked chassis system is also used but is less popular due to its more complicated design and higher price. In the case of a wheeled chassis system, the parameter for determining the amount of pressure generated on the soil is the area of contact between the tire and the soil—this parameter is affected by the stiffness of the tire, its size, and the inflation pressure and vertical load acting on the wheel. Scientific research on the impact of chassis systems on the soil can be divided into two main groups. The first includes experiments conducted under laboratory conditions in soil-filled cases or bins. These studies are not affected by atmospheric conditions and tend to present high repeatability of results (less risk of the influence of random factors). They also generally do not generate high costs for experiments. The disadvantages of laboratory studies include the prior preparation of soil for the experiment and space limitations in laboratory premises. Additionally, some of the factors occurring in real conditions (such as the presence of plant root mass) are not considered. The second group includes research conducted under real conditions, the main and most important advantage of which is obtaining conditions almost identical to those which occur during the operation of agricultural machinery under field conditions. Field research is characterized by greater variability in the results—primarily due to the significant influence of atmospheric conditions and natural processes that would not occur under laboratory conditions. Based on the test results, models are developed to describe the contact area between the tire and the soil. On the one hand, an accurate description of the shape of this area is made—most often super elliptical [7–10].

Sometimes, simplifications are made by introducing an alternate shape similar to a rectangle [11]. On the other hand, a very important result of research on tire–surface interaction is the development of mathematical models that allow the prediction of the parameters of the footprint or rut depending on operating parameters and external conditions [12].

A separate issue concerning the study of the impact of tires on the soil is the technique used when taking measurements. While the measurement of the length or width of the footprint does not present major difficulties, such difficulties may arise when determining the area of the footprint, especially since it is not on a single area (the footprint is three-dimensional). The simplest measurement techniques are based on planimetric techniques or analysis of photographs of the footprint. Diserens [13], Diserens et al. [14], and Taghavifar and Mardani [15] used in their studies an image-processing technique to determine the shape of the contact surface and the parameters affecting that shape. More advanced forms of research use computer techniques—such as the finite element method—to determine soil deformation. For example, González Cueto et al. [16], González Cueto et al. [17], Khot et al. [18], Nakashima and Kobayashi [19], and Smith et al. [20] used the finite element method to analyze tire–soil interaction. In contrast, the work of Nakashima and Oida [21] and Michael et al. [22] used a combination of the finite element method and the discrete element method to represent the model in the tire–soil system. Late experimenting, on the other hand, is based on the use of laser scanning of the footprint surface—such methods require sophisticated equipment and are costly, yet provide very high accuracy during surface representation and the ability to quickly read the basic dimensions of the footprint from the obtained scan [23,24].

Based on the literature review, it can be concluded that the continuous improvement in research methods for the tire–surface system is justified due to the minimization of negative impacts on the soil—primarily avoiding compaction. For the above reasons, this research aims to evaluate the impact of changes in selected operating parameters in tires of different designs on the footprint size of the soil under field conditions using 3D scanning techniques and digital image analysis. An additional goal is to verify the applicability of models developed from laboratory tests to predict soil deformation under field conditions.

## 2. Materials and Methods

This study was carried out under field conditions in an experimental field located at the Institute of Agricultural Engineering of Wrocław University of Environmental and Life

Sciences. The substrate was soil classified as sandy loam soil with a specific density of the soil skeleton of 2.59 g/cm$^3$. During the experiment using a bias-ply tire, the average soil moisture was 22.5%, and the average soil cone index was 0.59 MPa. In contrast, for the part of the experiment with a radial tire, the values of these parameters were 16% and 0.96 MPa. Soil parameters were measured using Eijkelkamp's Penetrologger, which included a ThetaProbe to measure the moisture content and a cone penetrometer to measure soil compactness (the apex angle of the measuring cone was equal to 60°, and the area of its base was equal to 0.0001 m$^2$). Two agricultural non-drive tires with different internal structures were tested: bias-ply (500/50-17) and radial (500/50R17). The tread type and tire dimensions were the same for both designs: width equal to 500 mm, profile height: 250 mm, rim diameter: 431.8 mm (17 inches), and outside diameter: 931.8 mm. The tires are designed for agricultural machinery such as straw/hay balers, manure spreaders, and forage trailers. Three tire pressures were adopted for the study: 0.8 bar, 1.6 bar, and 2.4 bar, and three values of vertical load on the tire: 7.8 kN, 11.8 kN, and 15.7 kN. The tests included measurement of tire footprint parameters on the soil: length, width, depth, and contact area.

*2.1. Test Bench*

The tire's footprint on the soil was created using the test bench by Ptak et al. (2023) in laboratory research, which required modification for field testing. The modification of the bench consisted mainly of installing elements that provided the possibility of aggregating the bench with a tractor—for this purpose, a three-point linkage was used. A schematic of the stand prepared for imprinting is shown in Figure 1. The vertical load on the tire was obtained using a hydraulic jack (5) which, for the duration of testing, was fitted in the vertical plane between the main frame (2) and the inner frame (3) of the test bench. The value of the vertical load was measured using a TecSis inductive dynamometer (4) with a measurement accuracy of 50 N and a measurement range of 0–100 kN. The tire being tested (1) was mounted on a shaft with bearings in the inner frame (3). Screw mechanisms (6) made it possible to lock the inner frame and its movement in the vertical plane and prevented pressure drops in the hydraulic jack, which could have resulted in an unintended reduction in the vertical load. With the use of weights (7), the test bench's load was achieved. No driving or braking forces were acting on the tire, which could have changed the stress distribution in the soil. After the footprint was created, the hydraulic jack was removed, the test bench with the tire being tested was raised using the tractor's three-point linkage, and then the tractor was driven off to specific locations in the field to create another footprint. The tire's footprint on the soil was created in five repetitions and its outline was marked, which was helpful in the further testing stage.

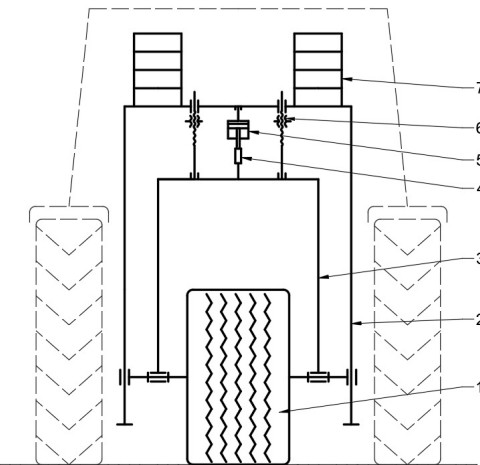

**Figure 1.** Scheme of the test bench: 1—wheel with tested tire, 2—main frame, 3—inner frame, 4—dynamometer, 5—hydraulic jack, 6—screw mechanism for locking the position of the inner frame, and 7—weights.

### 2.2. Scanning Process and Measurement of Footprint Parameters

Each time a footprint was created, it was scanned with a 3D scanner [25], the specifications of which are shown in Table 1. The scanner was connected to a laptop computer with Smarttech3Dmeasure software, which allowed the scanner to be operated. Due to the varying intensity of natural light, the scanning process was carried out in a tent, which ensured a uniform level of illumination of the scanned footprints.

**Table 1.** Technical data of the 3D scanner.

| Parameter | Description |
|---|---|
| Scanning technology | White structural light—LED |
| Measuring volume (x, y, z) (mm) | $400 \times 300 \times 240$ |
| Distance between points (mm) | 0.156 |
| Accuracy (mm) | 0.08 |

The scanning process resulted in a digital image of the footprint (in the form of a point cloud), which reflected its actual shape and size. The next step was to create a triangle mesh from the acquired point cloud and measure the parameters for later analysis. To read the depth of the footprint, creating a section in the vertical plane was necessary. Figure 2A shows the triangle mesh of the tire footprint and the intact portion of the soil that served as the reference point and the cross-section when measuring the depth of the footprint.

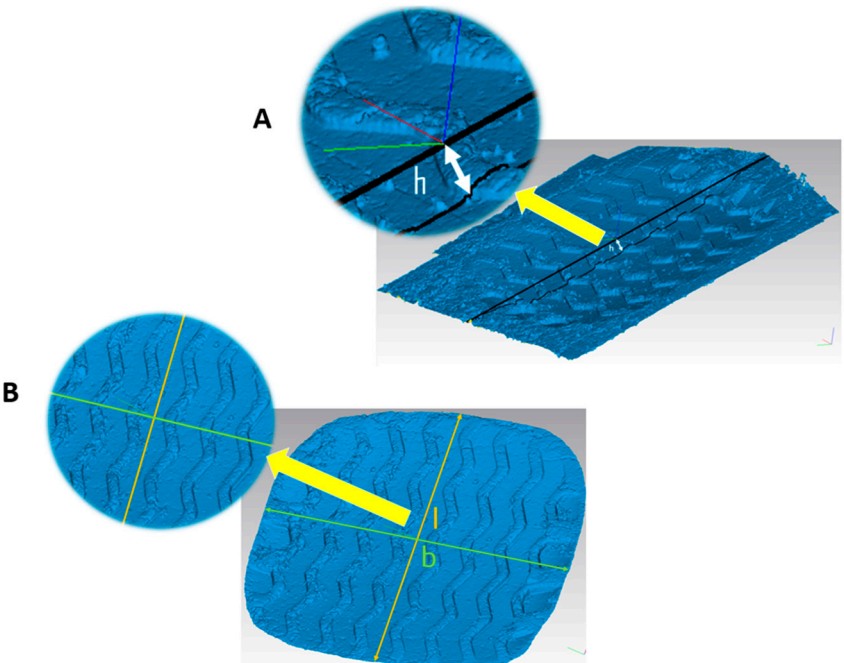

**Figure 2.** Mesh of triangles of the tire footprint on the soil: (**A**)—with intact soil (h—depth of the footprint); (**B**)—designated footprint (l—length of the footprint and b—width of the footprint).

The footprint depth was the height of the footprint section. Then, according to the markers surrounding the edge of the footprint, the intact soil part was removed from the image, and the next parameters analyzed in the experiment were determined—namely, the length of the footprint and its width, as shown in Figure 2B. It should be noted that the described parameters were always measured relative to the same measuring point located at the center of the footprint. The real surface was the surface of the footprint in three-dimensional space: not as a simplification in a two-dimensional projection, but as the whole footprint.

### 2.3. Statistical Analysis

Statistical analysis of the results obtained was carried out using Statistica 12.5. First, the results were subjected to analysis of variance at a significance level of $\alpha = 0.05$ with the separation of homogeneous groups (Fisher's NIR test). Subsequently, the suitability of the models developed for the data obtained in the laboratory part was verified.

## 3. Results

First, the linear dimensions of the footprint were evaluated. Figure 3 shows the results of measurements of the length and width of the footprints generated by the two tires tested.

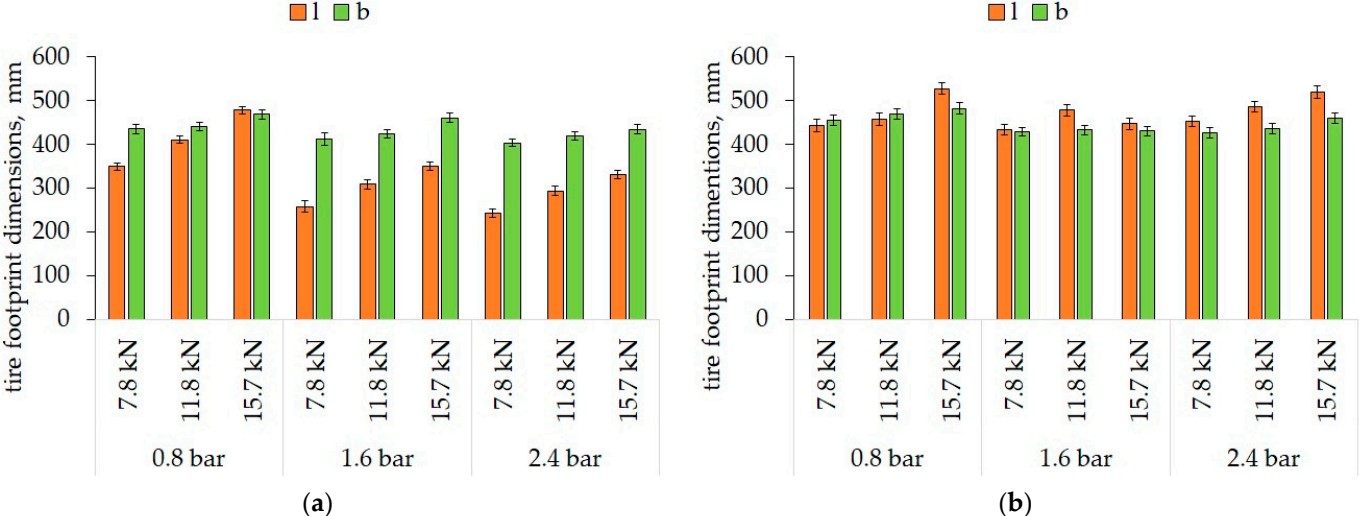

**Figure 3.** Values of tire footprint dimensions: radial tire (**a**); bias-ply tire (**b**); l—footprint length and b—footprint width.

In the case of the radial tire, an increase in the vertical load on the tire within the range of the same tire pressure always increased the measured parameters. The highest value of the radial tire's footprint length was 478.8 mm and was achieved at an inflation pressure of 0.8 bar and a vertical load of 15.7 kN. Reducing the vertical load to 11.8 kN and 7.8 kN at the lowest value of air pressure resulted in a decrease in the footprint length by 14% and 27%, accordingly. The lowest value of the footprint length generated by the radial tire was found at a tire pressure of 2.4 bar and a vertical load of 7.8 kN—it was 243.2 mm (49% less compared to the highest value observed for the measured parameter). For the bias-ply tire, the highest value of this parameter (528.2 mm) was observed at an inflation pressure of 0.8 bar and a vertical load of 15.7 kN, and a very similar value (519.8 mm) was found at a load of 15.7 kN and the highest inflation pressure (hence the conclusion that reducing the inflation pressure from 2.4 bar to 0.8 bar at the highest vertical load will result in an insignificant increase in footprint length—of only about 2%). The largest increase in the length of the bias-ply tire footprint with the increase in vertical load was observed at a pressure of 0.8 bar, increasing the vertical load from 11.8 kN to 15.7 kN (it was 16%).

Another analyzed parameter was the width of the footprint. The dynamics of change in this parameter were noticeably smaller than in the case of the footprint length. In the case of the radial tire, the highest value of the width parameter (468.9 mm) was observed at a tire pressure of 0.8 bar and a vertical load of 15.7 kN. By reducing the value of the vertical load to 11.8 kN and 7.8 kN in the range of the lowest inflation pressure, the width of the footprint decreased by 6% and 7%, respectively (to values of 440.5 mm and 435.4 mm, respectively). The largest increase in the measured parameter for the radial tire (8%) was noted at an inflation pressure of 1.6 bar and increasing the vertical load from 11.8 kN to 15.7 kN. At an inflation pressure of 2.4 bar and a load of 7.8 kN, the lowest value of the radial tire footprint width was observed, equal to 404.3 mm (14% less than the highest

value). For a vertical load of 11.8 kN, decreases in footprint width as a result of increasing inflation pressure were in the range of 4–5%, while at the highest vertical load level, changes in inflation pressure induced decreases in footprint width of 2% (an increase in inflation pressure from 0.8 bar to 1.6 bar) and 15% (an increase from 0.8 bar to 2.4 bar). For the bias-ply tire, the highest value of footprint width was observed, while for the radial tire, at an inflation pressure of 0.8 bar and a vertical load of 15.7 kN, a 3% increase (482.9 mm) in comparison with the radial tire. A bias-ply tire loaded at 7.8 kN at the inflation pressure of 2.4 bar generated a footprint width of 426.8 mm, corresponding to the smallest width observed for this tire (about 13% smaller than the largest footprint width). The largest increment (6%) in the current parameter was noticed at the highest inflation pressure level by increasing the vertical load from 11.8 kN to 15.7 kN. In other cases, the increments in footprint length were in the range of 2–3%. Evaluating the values of footprint width as a result of an increase in inflation pressure in the tire for one value of the vertical load, the largest decrease in the described parameter was observed at a vertical load of 15.7 kN and an increase in inflation pressure from 0.8 bar to 1.6 bar—the width of the footprint then decreased from 482.9 mm to 431.0 mm, respectively (a decrease of about 10%). Only a 0.5% decrease in the bias-ply tire's footprint width value was observed with a constant vertical load of 7.8 kN between the middle and highest adopted inflation pressures (429.0 mm to 426.8 mm).

Figure 4 shows the results of measurements of the next two parameters: depth and tire–soil contact area for the two tires tested.

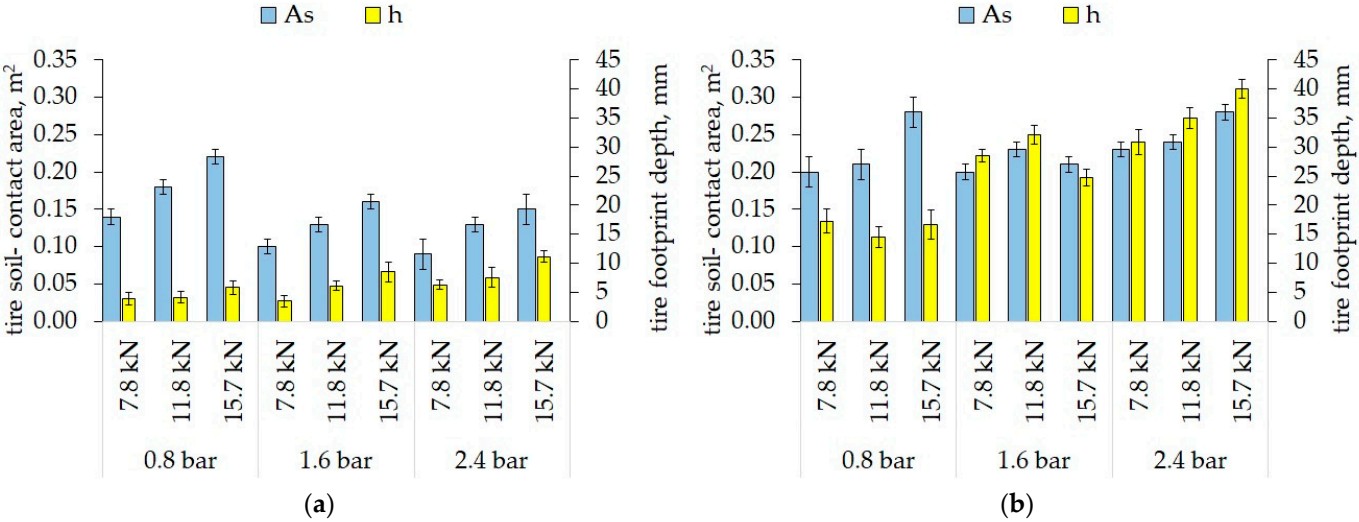

**Figure 4.** Values of tire footprint depth and tire–soil contact area: radial tire (**a**); bias-ply tire (**b**); $A_S$—tire–soil contact area and h—footprint depth.

Interpreting the footprint depth of the radial tire, it was noted that in all cases—at a constant value of inflation pressure—as the vertical load increased, the measured parameter increased as well. The smallest radial tire footprint depth (3.5 mm) was found at an inflation pressure of 1.6 bar and the smallest vertical load value. However, increasing the vertical load to 11.8 kN resulted in the greatest depth increase of 74%, resulting in a footprint depth of 6.1 mm. The highest value of the current parameter of 11.1 mm was found at an inflation pressure of 2.4 bar and the highest vertical load. The smallest increase in the current parameter for the radial tire was observed at an inflation pressure of 0.8 bar between the lowest and middle values of the vertical load—it was about 6%, which meant a change in the footprint depth from 3.9 mm to 4.1 mm. Inspecting the change in the value of the footprint depth due to an increase in tire pressure at the same vertical load, an increasing trend was observed in almost all cases. The largest increases in the parameter (48% and 46%) due to increasing the inflation pressure at a constant value of the vertical load were

observed with the first increase in pressure, i.e., from 0.8 bar to 1.6 bar for loads of 11.8 kN and 15.7 kN, respectively.

Analyzing the depth of bias-ply tire footprints, it was observed that in all cases, the footprints reached a depth of more than 10 mm. The smallest value of this parameter, at 14.6 mm, was observed at the lowest inflation pressure (0.8 bar) and at the middle value of the vertical load (11.8 kN). Reducing the vertical load to a value of 7.8 kN caused the bias-ply tire to generate a footprint that was 2.7 mm deeper (an increase of about 15% in the applied parameter). In turn, increasing the vertical load from the middle to the highest value also resulted in an increase in depth (from 14.6 mm to 16.6 mm, which corresponded to 14%). The opposite trend was observed at an inflation pressure of 1.6 bar. At the lowest vertical load, the depth of the footprint was equal to 28.5 mm, and an increase in the vertical load resulted in an increase in depth of about 13% (to a value of 32.2 mm), while as a result of another increase in load (to 15.8 kN), the depth decreased to 24.7 mm. On the other hand, at the highest adopted inflation pressure, the trend of the change in the footprint depth was opposite to the lower inflation pressures. The first increase in the vertical load on the tire to 11.8 kN resulted in an increase in the footprint depth of about 13% to a value of 35.0 mm, while the subsequent increase in the vertical load (to 15.7 kN) resulted in an increase in the footprint depth to a value of 40.0 mm, which was also the greatest footprint depth for a bias-ply tire.

The last parameter analyzed was the contact area between the tire and the soil. The highest value of this parameter for a radial tire ($0.22 \text{ m}^2$) was observed at the lowest inflation pressure and the highest vertical load (0.8 bar and 15.7 kN, respectively). It is noteworthy that in the 0.8 bar inflation pressure, reducing the vertical load to 11.8 kN and then to 7.8 kN resulted in a decrease in the tire–soil contact area each time by a constant $0.04 \text{ m}^2$ to values of $0.18 \text{ m}^2$ and $0.14 \text{ m}^2$ (decreases of 22% and 29%, respectively). A similar trend was observed in the inflation pressure of 1.6 bar, where a reduction in vertical load from 15.7 kN to 11.8 kN and 7.8 kN resulted in a reduction in the tire–soil contact area by $0.03 \text{ m}^2$ (decreases of 23% and 30%). At the highest vertical load, the tire–soil contact area was equal to $0.16 \text{ m}^2$, and with the change in vertical load to lower values, it decreased by $0.03 \text{ m}^2$ to values of $0.13 \text{ m}^2$ and $0.10 \text{ m}^2$, respectively. The smallest value ($0.09 \text{ m}^2$) of the current parameter was found at an inflation pressure of 2.4 bar and a vertical load of 7.8 kN.

The values of the contact area between the bias-ply tire and the soil were significantly larger compared to the radial tire (at the same inflation pressures and vertical loads). The smallest value of the inspected parameter for the bias-ply tire equal to $0.20 \text{ m}^2$ was observed at a vertical load of 7.8 kN and the lowest and middle inflation pressure. At an inflation pressure of 0.8 bar, an increase in the vertical load from 7.8 kN to 11.8 kN resulted in a 5% increase in the analyzed parameter. In contrast, a further increase in vertical load resulted in a 33% increase in the contact area to a value of $0.28 \text{ m}^2$, the largest observed value of the tire–soil contact area. A value of $0.28 \text{ m}^2$ was also observed at the highest tire pressure (2.4 bar) and the highest vertical load (15.7 kN). For this pressure, reducing the vertical load to 11.8 kN and then to 7.8 kN resulted in a 14% and 4% decrease in the value of the contact area (to values of $0.24 \text{ m}^2$ and $0.23 \text{ m}^2$), respectively. In the case of inflation pressure at 1.6 bar, a change in vertical load from the lowest to 11.8 kN resulted in a larger contact area (an increase from $0.20 \text{ m}^2$ to $0.23 \text{ m}^2$, i.e., by 15%), but a subsequent increase in load resulted in a 9% decrease in the current parameter (to a value of $0.21 \text{ m}^2$).

### 3.1. Statistical Analysis Results

#### 3.1.1. Analysis of Variance—Results for Radial Tire

Table 2 shows the results of the multiple analysis of variance for the results obtained for the radial tire. The *p*-values shown indicate the probability of accepting the null hypothesis representing the absence of a significant effect of a factor on the analyzed parameter.

**Table 2.** The statistical analysis of experimental data for the radial tire; the significance level $\alpha = 0.05$, and SD—standard deviation.

| Footprint Parameter | Factor | Factor Level | Arithmetic Mean | ±SD | *p*-Value |
|---|---|---|---|---|---|
| Width of the footprint (b), mm | Vertical load | 7.8 kN | 417.7 [A] | 16.98 | <0.00001 |
| | | 11.8 kN | 428.5 [B] | 12.63 | |
| | | 15.7 kN | 455.0 [C] | 18.04 | |
| | Inflation pressure | 0.8 bar | 448.3 [A] | 17.81 | 0.00004 |
| | | 1.6 bar | 433.3 [B] | 23.80 | |
| | | 2.4 bar | 419.5 [C] | 15.92 | |
| Length of the footprint (l), mm | Vertical load | 7.8 kN | 283.9 [A] | 50.82 | <0.00001 |
| | | 11.8 kN | 338.1 [B] | 55.77 | |
| | | 15.7 kN | 387.2 [C] | 69.64 | |
| | Inflation pressure | 0.8 bar | 413.3 [A] | 56.29 | <0.00001 |
| | | 1.6 bar | 306.4 [B] | 41.18 | |
| | | 2.4 bar | 289.6 [C] | 39.53 | |
| Depth of the footprint (h), mm | Vertical load | 7.8 kN | 4.53 [A] | 1.53 | <0.00001 |
| | | 11.8 kN | 5.90 [B] | 1.86 | |
| | | 15.7 kN | 8.47 [C] | 2.58 | |
| | Inflation pressure | 0.8 bar | 4.59 [A] | 1.28 | 0.00001 |
| | | 1.6 bar | 6.00 [B] | 2.42 | |
| | | 2.4 bar | 8.31 [C] | 2.45 | |
| Tire–soil contact area ($A_s$), m² | Vertical load | 7.8 kN | 0.11 [A] | 0.03 | <0.00001 |
| | | 11.8 kN | 0.15 [B] | 0.03 | |
| | | 15.7 kN | 0.18 [C] | 0.04 | |
| | Inflation pressure | 0.8 bar | 0.18 [A] | 0.04 | <0.00001 |
| | | 1.6 bar | 0.13 [B] | 0.03 | |
| | | 2.4 bar | 0.12 [B] | 0.03 | |

The letters at the arithmetic mean ([A], [B], and [C]) denote separate homogenous groups.

According to the data presented in the table above, it can be concluded that both factors had a significant effect on all four analyzed parameters, as the *p*-values were significantly lower than the accepted significance level α. Reading the results of the post hoc tests, it was noted that only in the case of the effect of tire inflation pressure on the footprint area were the two levels (1.6 bar and 2.4 bar) classified into one group, which meant that there were no significant differences between these levels in terms of the footprint area. In all other cases, each factor level was classified as a separate homogeneous group.

As part of the statistical analysis, a model describing the relationship between the operating factors and the actual footprint area was verified. This model was developed in the laboratory part of the experiment and is presented in detail by Ptak et al. [26]. Its general form is shown in Equation (1).

$$A_s = -0.267 - 0.0427 \cdot \ln p_i + 0.012 \left( \ln \frac{G}{0.00981} \right)^2 - 0.0035 \left( \frac{G}{0.00981} \right)^{0.5} \tag{1}$$

where:

$A_s$—contact area of the footprint (m²);

$p_i$—inflation pressure (bar);

$G$—vertical load (kN).

Verification of the model's suitability was carried out by comparing the experimental data (from the field part of the experiment) with the data calculated from the model. A linear regression analysis (using the F-Fisher test at a significance level of $\alpha = 0.05$) was performed for the resulting data. Figure 5 shows the relationship between the two groups of numerical data.

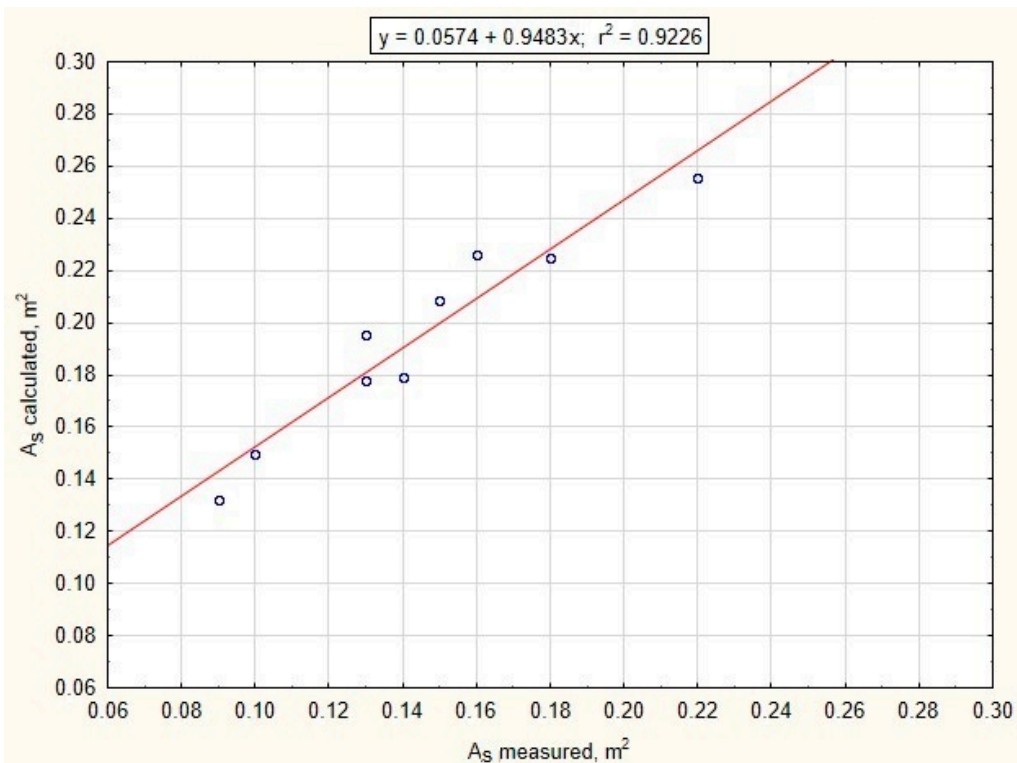

**Figure 5.** Relationship between the area values calculated from the model and the results of measurements under field conditions for footprints generated by the radial tire; $A_S$—area of the footprint.

Evaluating the comparison shown in Figure 5, one can see that the clustering of points around the straight line was high. The obtained value of the determination coefficient was 0.9226, which indicates a good fit of the model to the measurement data. Thus, it can be concluded that in the case of a radial tire, the model based on laboratory tests allows forecasting the contact area under field conditions (with known values of vertical load and inflation pressure). This thesis was also confirmed after testing the significance of the regression with the F-Fisher test. In this testing, the null hypothesis of non-significance of the regression coefficient and slope was assumed. The obtained value of the test function was equal to F(1,7) = 83.49, and the probability level of accepting the null hypothesis was equal to $p = 0.00004$. Since the probability level was much lower than the assumed significance level (0.05) and the value of the test function was large, the null hypothesis was rejected—this meant that the correlation coefficient was significant. The slope was also significant ($p$-value = 0.0076).

### 3.1.2. Analysis of Variance—Results for Bias-Ply Tire

Table 3 shows the results of the multivariate analysis of variance for the results obtained for the bias-ply tire. As in the previous case, the $p$-values denote the probability of accepting the null hypothesis representing the absence of a significant effect of a factor on the analyzed parameter.



**Table 3.** The statistical analysis of experimental data for the bias-ply tire; the significance level $\alpha = 0.05$, and SD—standard deviation.

| Footprint Parameter | Factor | Factor Level | Arithmetic Mean | ±SD | *p*-Value |
|---|---|---|---|---|---|
| Width of the footprint (b), mm | Vertical load | 7.8 kN | 437.3 [A] | 17.34 | 0.00726 |
| | | 11.8 kN | 446.2 [A] | 20.04 | |
| | | 15.7 kN | 458.0 [B] | 24.90 | |
| | Inflation pressure | 0.8 bar | 469.4 [A] | 15.63 | <0.00001 |
| | | 1.6 bar | 431.1 [B] | 9.51 | |
| | | 2.4 bar | 440.9 [B] | 18.40 | |
| Length of the footprint (l), mm | Vertical load | 7.8 kN | 443.7 [A] | 14.07 | 0.00019 |
| | | 11.8 kN | 474.1 [B] | 17.36 | |
| | | 15.7 kN | 498.7 [C] | 39.89 | |
| | Inflation pressure | 0.8 bar | 476.4 [A] | 41.03 | 0.01856 |
| | | 1.6 bar | 453.7 [B] | 22.86 | |
| | | 2.4 bar | 486.5 [A] | 30.91 | |
| Depth of the footprint (h), mm | Vertical load | 7.8 kN | 25.55 [A] | 6.50 | 0.51237 |
| | | 11.8 kN | 27.24 [A] | 9.71 | |
| | | 15.7 kN | 27.25 [A] | 10.57 | |
| | Inflation pressure | 0.8 bar | 16.15 [A] | 2.22 | <0.00001 |
| | | 1.6 bar | 28.48 [B] | 3.45 | |
| | | 2.4 bar | 35.40 [C] | 4.43 | |
| Tire–soil contact area (A$_s$), m$^2$ | Vertical load | 7.8 kN | 0.21 [A] | 0.02 | 0.00063 |
| | | 11.8 kN | 0.23 [A] | 0.02 | |
| | | 15.7 kN | 0.26 [B] | 0.04 | |
| | Inflation pressure | 0.8 bar | 0.23 [A] | 0.04 | 0.00672 |
| | | 1.6 bar | 0.21 [A] | 0.02 | |
| | | 2.4 bar | 0.25 [B] | 0.02 | |

The letters at the arithmetic mean ([A], [B], and [C]) denote separate homogenous groups.

Inspecting the data presented in the table above, it can be concluded that the tire inflation pressure had a significant effect on all the analyzed parameters. In contrast, the second factor (vertical load) was found to have no significant effect on the depth of the footprint generated (in the other analyzed parameters, the significance was maintained). In addition, based on the results of post hoc tests (Fisher's NIR), it was found that in many cases, adjacent levels of the factor were qualified to the same group (e.g., levels of 1.6 bar and 2.4 bar when analyzing the effect of pressure on the width of the footprint, or levels of 7.8 kN and 11.8 kN when evaluating the effect of the load on the footprint area).

Verification of the correctness of the model selection was also carried out for the footprints generated by a bias-ply tire. As in the case of the radial tire, a model was developed based on laboratory tests presented in Ptak et al. [26]. The model had the following form (Equation (2)):

$$A_s = 0.947 - \frac{1.882}{p_i{}^{0.5}} + \frac{1.063}{p_i} + 7.76 \cdot 10^{-4} G \cdot \ln \frac{G}{0.00981} \tag{2}$$

where:

$A_s$—contact area of the footprint (m$^2$);

$p_i$—inflation pressure (bar);

$G$—vertical load (kN).

Figure 6 compares the data calculated from the model with the data acquired during field measurements. As in the case of the model for the radial tire, regression analysis was carried out using the F-Fisher test at a significance level of $\alpha = 0.05$.

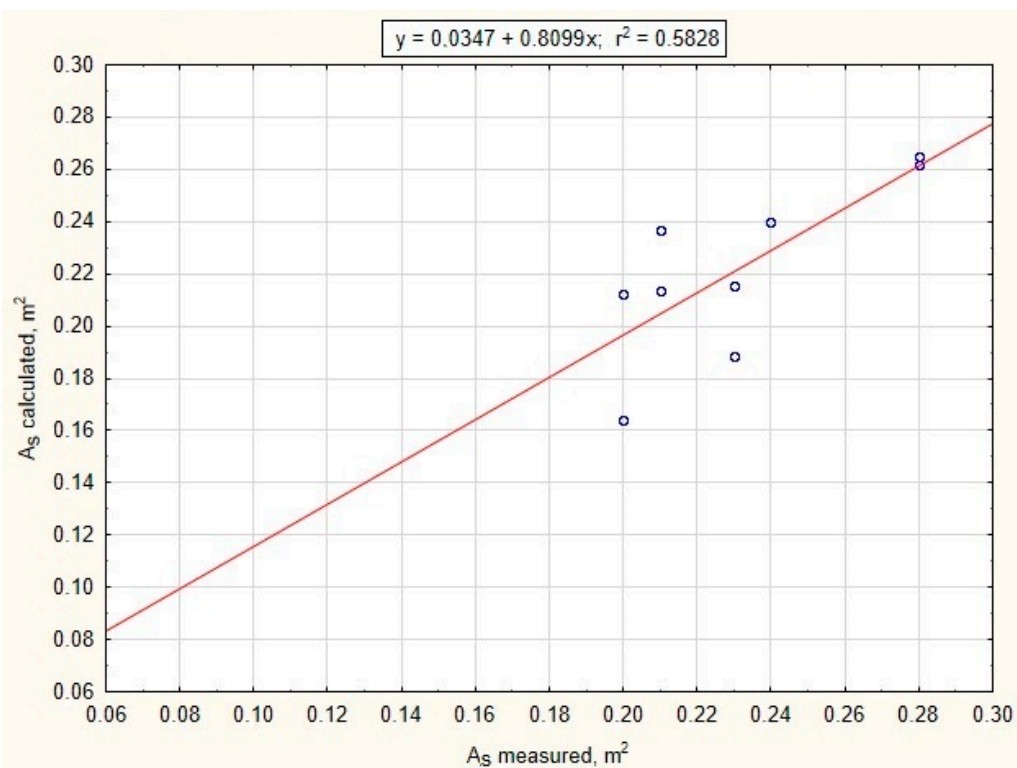

**Figure 6.** Relationship between the area values calculated from the model and the results of measurements under field conditions for footprints generated by a bias-ply tire; $A_S$—footprint area.

In the case of the bias-ply tire, a poorer fit between the data calculated from the model and the actual data is evident—there is a greater scattering of points from the regression line. The significance testing of the regression coefficient with the F-Fisher test showed that the value of the test function was equal to $F(1,7) = 9.778$, and the probability level of accepting the null hypothesis was equal to $p = 0.0167$. This meant that the determination coefficient was considered significant. However, the significance of the slope was not demonstrated (probability level $p = 0.584$). Combined with the relatively low determination coefficient ($R^2 = 0.58$), it can be concluded that the model developed for the bias-ply tire based on laboratory tests has limited applicability under field conditions (only 58% of the model results would coincide with actual results). To better reflect the data, developing a model that would consider additional parameters related to, for example, tire stiffness and parameters describing the ground condition is recommended.

## 4. Discussion

The experiment conducted under field conditions made it possible to demonstrate differences in the dimensions of the soil footprint created by tires of different internal designs but with the same external dimensions at different vertical wheel loads and inflation pressures. Based on the analysis of the results, the adopted values of the experiment's variable factors differently affect the measured parameters, including the length and width of the footprint; the depth; and, in particular, the area of contact between the tire and the soil. In addition, the suitability of models developed based on laboratory tests for predicting the size of the footprint area under field conditions was evaluated.

Many authors indicate that a reduction in inflation pressure has the effect of increasing tire–soil contact area [27–29]. Diserens [13], examining a range of tires, showed a divergent effect of tire inflation pressure on the tire–soil contact area. Yadav and Raheman [30] showed that the tire–soil contact area increased with the vertical load on the tire. In addition, the rate of increase of the analyzed parameter decreased with an increase in tire size. It was also found that the tire–soil contact area decreased exponentially with an increase in inflation

pressure from 69 kPa to 234 kPa for any vertical tire load. Similar results were observed for both bias-ply and radial tires in the work of Sharma and Pandey [31], Schjønning et al. [27], Schjønning et al. [32], and Kumar et al. [33].

Thus, it seems reasonable to analyze the tire–soil contact surface as a three-dimensional image, also taking into account the depth of the rut formed by the tire. This is because analysis of the footprint as a plane can lead to oversimplifications, and the error will be greater in the depth of the footprint. Studies taking into account this parameter and the amount of soil deformation were conducted by Kenarsari et al. [23] and Pierzchała et al. [34]. This study showed that a similar value of the contact area of radial and bias-ply tires with the soil was obtained at a much greater footprint depth of the bias-ply tire. In the inflation pressure of 0.8 bar, a contact area in the range of 0.21 $m^2$–0.22 $m^2$ was observed with a vertical load of 15.7 kN and 11.8 kN, and the footprint depth for these parameters was equal to 5.8 mm and 14.6 mm (for the radial and bias tire, respectively).

In this case, the footprint depth of the bias-ply tire was about 150% greater compared to the footprint of the other tire tested. The vertical load of the bias-ply tire with a value of 15.7 kN but at an inflation pressure of 1.6 bar resulted in a tire–soil contact area equal to 0.21 $m^2$, but the footprint depth was 24.7 mm. It should be noted that a similar value of tire–soil contact area for the tires tested was observed at the same vertical load (15.7 kN) but at a lower pressure for the radial tire (0.8 bar). As a result, the bias-ply tire generated a footprint of 328%, making it more destructive to the soil than the radial tire's footprint. At an inflation pressure of 1.6 bar, loading the radial tire with the highest accepted load value, a tire–soil contact area of 0.16 $m^2$ was observed. At the same inflation pressure but with the lowest vertical load, the bias-ply tire had a soil contact area of 0.20 $m^2$. This value was only 0.04 $m^2$ greater compared to the radial tire, but the bias-ply tire's footprint depth was over 20 mm greater at the same time.

Verifying the validity of the use of models describing the tire footprint area under field conditions, it can be concluded that only in the case of a radial tire can the developed model predict the contact area under real conditions. On the other hand, there are significant limitations to the use of the model developed under laboratory conditions for the bias-ply tire; the model could be used to a limited extent to predict the footprint area under real conditions (only 58% of the real results would agree with the results calculated from the model). To better fit the model to the real data, additional factors would need to be considered—primarily those related to the condition of the soil. This thesis is confirmed in the literature, where models of footprint surface and other parameters related to tire interaction with the soil are discussed [12].

## 5. Conclusions

Based on the conducted tests, it was found that:

1. Unambiguous determination of the factors affecting the tire–soil contact area is difficult due to their diversity. It should also be emphasized that the tire–soil contact area is the overriding parameter responsible for the distribution of stresses in the soil. It is therefore necessary to continue research in this direction, as this will facilitate the selection of appropriate operating parameters of agricultural tires for soil conditions. By verifying the models describing the tire footprint under laboratory conditions, it was found reasonable to use the model under field conditions, but only for a radial tire. In the case of a bias-ply tire, only 58% of the actual results agreed with those calculated from the model.

2. In the range of the same value of inflation pressure, the length and width of the radial tire footprint, in all cases, increased with increasing vertical load. Similarly, at a constant vertical load, a decrease in the discussed parameters was observed with an increase in inflation pressure. In the case of the bias-ply tire footprint, the trend was similar, except for an inflation pressure of 1.6 bar (the first increase in vertical load resulted in a decrease in the length and width of the footprint, and the next again caused an increase in the measured parameters).

3.  The footprint depth and contact area of the radial tire increased with increasing vertical load at constant inflation pressure. As with the bias-ply tire's footprint length and width, its footprint depth and contact area increased due to increasing the vertical load at constant inflation pressure. The exception was a tire pressure range of 1.6 bar, at which the first increase in vertical load increased the footprint depth and the area of contact between the tire and the soil, while subsequent increases resulted in a decrease in these parameters.

**Author Contributions:** Conceptualization, J.C. and W.P.; methodology, J.C.; software, M.B. and W.P.; validation, M.B. and A.M.; formal analysis, J.C. and K.L.; investigation, W.P., M.B. and A.M.; resources, W.P.; data curation, M.B.; writing—original draft preparation, W.P. and M.B.; writing—review and editing, M.B.; visualization, W.P.; supervision, J.C.; project administration, J.C. and K.L.; funding acquisition, J.C. and K.L. All authors have read and agreed to the published version of the manuscript.

**Funding:** The article processing charge was financed by the Wroclaw University of Environmental and Life Sciences.

**Institutional Review Board Statement:** Not applicable.

**Data Availability Statement:** Not applicable.

**Conflicts of Interest:** The authors declare no conflict of interest. The funders had no role in the design of the study; in the collection, analyses, or interpretation of data; in the writing of the manuscript; or in the decision to publish the results.

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
