# Peer review of "Evaluation of Tires Acting on Soil in Field Conditions Using the 3D Scanning Method"

_agriculture, doi:10.3390/agriculture13051094_

Round 1
Reviewer 1 Report
This paper presents some very interesting data, but it really needs a very substantial edit to make the English usage better and improve readability. I think the authors also need to remember not to present the same data both graphically and in the text. The text of the results includes considerable reiteration and discussion of data presented in charts, which could be presented more briefly in the discussion of results.
Find these results generated many questions. The radial ply tire is generally expected to provide a longer footprint – here it was shorter. It also provided a smaller contact area, and one which more closely approximated the elementary idea of load = pressure x area. A small area with reduced penetration must surely reflect more uniform pressure distribution within the footprint and the greater flexibility of the radial ply carcass.
Alternatively, the reverse of the above applies to the bias ply tire, reflecting the influence of the stronger carcass, and I was quite surprised that the discussion did not include these aspects, or the possible differences between static and dynamic situations.
This paper needs a thorough edit for English usage. I started the process (by line number) below, but there is a major task for someone here.
Line No.
23 less accurate prediction of the surface area under field conditions on the basis of laboratory tests.
38 suggest replacing this sentence with “Heavy field traffic can cause soil compaction”
87 the late experiment? Perhaps “more recent techniques use laser scanning………….
104 skeleton? Perhaps soil particle density
110 drive tires, not tractor tires? But no make or designation or details of construction? These things are important because the circumferential belt has a very significant effect on radial tire performance. The belt might not be present in a non-drive tire with no traction lugs.
121 suggest “This test was modified to allow its installation on a tractor three-point linkage”
133 this is just a static load? I suspect shape would be different in the dynamic situation.
Figure 3 be good to see tick marks on the vertical axes (this and other figures)
179 suggest “dimensions: (a) radial tire; (b) bias-ply tire…….
Author Response
Dear Reviewer,
Thank you for your comments and suggestions. We are so glad that you concluded that our publication is up-to-date. According to your suggestion, the manuscript was checked and corrected by MDPI system. Answers to your questions you can find below.
Sincerely,
Authors
1.This paper presents some very interesting data, but it really needs a very substantial edit to make the English usage better and improve readability. I think the authors also need to remember not to present the same data both graphically and in the text. The text of the results includes considerable reiteration and discussion of data presented in charts, which could be presented more briefly in the discussion of results.
Answer: The final version of the manuscript was corrected by English Editor from MDPI. We also corrected the “Results” section – now the most important issues (selected values of parameters, significant changes in values) are presented in this section.
2. Find these results generated many questions. The radial ply tire is generally expected to provide a longer footprint – here it was shorter. It also provided a smaller contact area, and one which more closely approximated the elementary idea of load = pressure x area. A small area with reduced penetration must surely reflect more uniform pressure distribution within the footprint and the greater flexibility of the radial ply carcass.
3. Alternatively, the reverse of the above applies to the bias ply tire, reflecting the influence of the stronger carcass, and I was quite surprised that the discussion did not include these aspects, or the possible differences between static and dynamic situations.
Answers to comments 2 and 3: In the case of our experiment, the weather conditions were variable. In fact, this reflected real situations at the agricultural operations – for this reason the conditions of exploitation of agricultural wheel can be differ. However, variable conditions I our experiment affected soil moisture. Radial tire was tested on the soil with lower moisture than in the case of bias-ply tires – these differences were describe in “Methodology” section. In turn, it can be significant factor affected the deformability of the soil during the footprint generating. This difference was confirmed by differences in soil compaction values. Probably, for these reasons the bias-ply tire generated footprint area with higher area than radial tire.
4. This paper needs a thorough edit for English usage. I started the process (by line number) below, but there is a major task for someone here.
Line No.
23 less accurate prediction of the surface area under field conditions on the basis of laboratory tests.
38 suggest replacing this sentence with “Heavy field traffic can cause soil compaction”
87 the late experiment? Perhaps “more recent techniques use laser scanning………….
104 skeleton? Perhaps soil particle density
110 drive tires, not tractor tires? But no make or designation or details of construction? These things are important because the circumferential belt has a very significant effect on radial tire performance. The belt might not be present in a non-drive tire with no traction lugs.
121 suggest “This test was modified to allow its installation on a tractor three-point linkage”
133 this is just a static load? I suspect shape would be different in the dynamic situation.
Figure 3 be good to see tick marks on the vertical axes (this and other figures)
179 suggest “dimensions: (a) radial tire; (b) bias-ply tire…….
Answer: According to your suggestion, the manuscript has undergone English language editing by MDPI and figures was corrected.
Reviewer 2 Report
The article submitted for review is interesting, from a scientific point of view, well conceived and structured, however, a careful reading of it reveals the following issues that should be resolved by the authors:
- the description of the stand is not adequate, only a simplified scheme is presented which is not functional and from which the role of the hydraulic jack is not seen. The same problem was observed in the bibliographic reference Ptak, 2022. In order to remedy this observation, I recommend the authors to redo the stand diagram and insert photos from the time of the scientific approach;
- no information is given on the history of the crops previously cultivated, and whether the farming system is conventional (ploughing), minimum tillage or no-till;
- it is not specified whether the trials were carried out in a static or dynamic system.
Author Response
Dear Reviewer
Thank you for your comments and suggestions. We are so glad that you concluded that our publication is up-to-date. According to your suggestions and questions, answers to your comments you can find below.
Sincerely
Authors
1. The description of the stand is not adequate, only a simplified scheme is presented which is not functional and from which the role of the hydraulic jack is not seen. The same problem was observed in the bibliographic reference Ptak, 2022. In order to remedy this observation, I recommend the authors to redo the stand diagram and insert photos from the time of the scientific approach;
Answer: Hydraulic jack was used to obtain differ values of vertical load acting on the wheel. It was mounted between inner frame and main frame. As a result of pressure increase and sliding out of the piston, vertical acting load was pressed on inner frame, then it was transferred on the wheel with tested tire. An increase in working pressure in hydraulic jack cylinder resulted in an increase in wheel load - thus them bigger vertical load acting on the tire. The weights were mounted on the main frame in order to stabilize only (during the experiment their weight was constant), to prevent uncontrollable move the test bench along vertical plane. Stabilization along horizontal plane was assured using stiff connection with three-point linkage of tractor.
2. No information is given on the history of the crops previously cultivated, and whether the farming system is conventional (ploughing), minimum tillage or no-till.
Answer: In the last year before starting the experiment, no plants were grown on the soil used in the research. Before starting the main part of experiment, the soil has been loosened using a cultivator and them compacted using smooth roller to value of compaction equal 0.59 MPa and 0.96 MPa for bias- ply tire and radial tire, respectively.
3. It is not specified whether the trials were carried out in a static or dynamic system.
Answer: The experiment was conducted in static conditions. The wheel was just placed in designated place on tested field to generate tire footprint in soil, the wheel wasn’t turn on soil; the tires used in experiment were non driving tires: there were no driving or braking forces acting on wheel. It was caused by fact, that it was the first stage of experiment. It is necessary to recognize the phenomena at the contact of the tire, before its is moving, with soil. Described experiment is one of stages of the whole research concerning „tire-surface” system; continuation of research will be also conducted in dynamic conditions.
Round 2
Reviewer 1 Report
The authors rightly point out that they had recorded the different conditions for the two tests (which I didn't notice). Good that they did so, but when assessments are made under different conditions is obviously reduces the validity of the comparison – in this case between bias and radial ply tires. One could really expect this to be done side-by-side with some replication).
This is still useful as a techniques paper, but the results are less useful than I had originally thought. I'm also surprised that it is being published in an agriculture journal rather than one focusing on soil management.
This is not easy to read, but in most cases it is not too difficult to work out what the authors are trying to say. I still think it would repay a thorough edit , but its is useful as a techniques paper.
All I'm also surprised that it is being published in an agriculture journal rather than one focusing on soil management.